# Trials and Tribulations of MicroRNA Therapeutics

**DOI:** 10.3390/ijms25031469

**Published:** 2024-01-25

**Authors:** Attila A. Seyhan

**Affiliations:** 1Laboratory of Translational Oncology and Experimental Cancer Therapeutics, Warren Alpert Medical School, Brown University, Providence, RI 02912, USA; attila_seyhan@brown.edu; 2Department of Pathology and Laboratory Medicine, Warren Alpert Medical School, Brown University, Providence, RI 02912, USA; 3Joint Program in Cancer Biology, Lifespan Health System and Brown University, Providence, RI 02912, USA; 4Legorreta Cancer Center, Brown University, Providence, RI 02912, USA

**Keywords:** microRNAs, miRNAs, post-transcriptional gene regulation, miRNA therapeutics, miRNA mimics, antimirs, antagomirs

## Abstract

The discovery of the link between microRNAs (miRNAs) and a myriad of human diseases, particularly various cancer types, has generated significant interest in exploring their potential as a novel class of drugs. This has led to substantial investments in interdisciplinary research fields such as biology, chemistry, and medical science for the development of miRNA-based therapies. Furthermore, the recent global success of SARS-CoV-2 mRNA vaccines against the COVID-19 pandemic has further revitalized interest in RNA-based immunotherapies, including miRNA-based approaches to cancer treatment. Consequently, RNA therapeutics have emerged as highly adaptable and modular options for cancer therapy. Moreover, advancements in RNA chemistry and delivery methods have been pivotal in shaping the landscape of RNA-based immunotherapy, including miRNA-based approaches. Consequently, the biotechnology and pharmaceutical industry has witnessed a resurgence of interest in incorporating RNA-based immunotherapies and miRNA therapeutics into their development programs. Despite substantial progress in preclinical research, the field of miRNA-based therapeutics remains in its early stages, with only a few progressing to clinical development, none reaching phase III clinical trials or being approved by the US Food and Drug Administration (FDA), and several facing termination due to toxicity issues. These setbacks highlight existing challenges that must be addressed for the broad clinical application of miRNA-based therapeutics. Key challenges include establishing miRNA sensitivity, specificity, and selectivity towards their intended targets, mitigating immunogenic reactions and off-target effects, developing enhanced methods for targeted delivery, and determining optimal dosing for therapeutic efficacy while minimizing side effects. Additionally, the limited understanding of the precise functions of miRNAs limits their clinical utilization. Moreover, for miRNAs to be viable for cancer treatment, they must be technically and economically feasible for the widespread adoption of RNA therapies. As a result, a thorough risk evaluation of miRNA therapeutics is crucial to minimize off-target effects, prevent overdosing, and address various other issues. Nevertheless, the therapeutic potential of miRNAs for various diseases is evident, and future investigations are essential to determine their applicability in clinical settings.

## 1. Introduction

While most of the research in oncology predominantly centers around the ever-changing aspects of proteins and the RNA molecules responsible for coding those proteins, it is important to note that these coding sequences account for only about 2% of the genome (https://www.genomicseducation.hee.nhs.uk/genotes/knowledge-hub/non-coding-dna/) (accessed on 20 December 2023) [1,2,3]. However, the remaining 98% of the genome, which includes non-coding RNAs (ncRNAs) such as microRNAs (miRNAs), plays pivotal roles in numerous biological processes during normal physiological processes, as well as in the onset and development of various diseases, including various types of human cancers [4]. This emphasizes the significance of miRNAs and other non-coding RNAs in the initiation and progression of tumors.

In addition, miRNAs play key functions in the modulating expression of numerous genes both at the transcriptional [5,6,7] and post-transcriptional [8,9,10,11] levels, and exhibit tissue-specific [12,13] and developmental expression patterns [14,15,16], showcasing their significance in a diverse range of biological processes within cells and organisms. Altered expression of miRNAs has emerged as an additional molecular mechanism implicated in the pathogenesis of numerous diseases [17,18,19], spanning innate immunity [20], autoimmunity and autoimmune diseases [21], viral infections [22,23,24,25], acute hepatitis [26], depression [27], anxiety [28], Alzheimer’s disease [29], Huntington’s disease [30], metabolic and cardiovascular diseases [31,32,33,34], diabetes [8,33,34,35,36,37,38], and many types of cancers [12,39,40,41,42,43,44,45,46,47,48,49,50,51,52,53,54,55,56,57,58,59,60,61,62,63,64,65,66,67,68,69]. Consequently, these miRNAs can serve as indicators for the presence of a pathological condition, as well as provide insights into its stage, progression, or genetic associations.

More recently, there is emerging evidence suggesting that diet-derived exogenous miRNAs (or “xenomiRs”) can enter the circulatory system and tissues, potentially influencing gene expression and biological functions [70,71,72,73,74,75].

The absorption of miRNAs by gastric and intestinal cells, along with their potential impact on the gut microbiota and their potential immunomodulatory effects suggests the potential for cross-species or cross-kingdom communication via miRNAs [75]. Because of these observations, one potential method of administering miRNAs is orally.

miRNAs are often associated with extracellular vesicles (EVs), RNA-binding proteins, lipoproteins, or lipid derivatives, along with nanoparticles.

These protective elements shield miRNAs against the harsh gastrointestinal environment, which encompasses salivary and pancreatic RNases, the low pH of the stomach, digestive enzymes, peristaltic activity, and microbial enzymes. This protective shield presumably aids in the absorption of miRNAs from the digestive tract [75].

However, there is ongoing debate surrounding the absorption, stability, and physiological impact of these food-derived miRNAs. Conflicting findings exist regarding the bioavailability and the functional role of plant food-contained miRNAs in human systems [76,77].

Ongoing research continues to uncover new insights into the molecular mechanisms that drive the dysregulation of miRNA biogenesis and aberrant expression in cancer.

For example, it is widely recognized that various factors such as genetic deletions or amplifications, epigenetic methylation of miRNA genomic loci, and modifications affecting the regulation of primary miRNAs (pri-miRNA) by transcription factors, alongside components involved in the miRNA biogenesis pathway frequently lead to alterations in miRNA expression and function across numerous cancer types [56,78,79].

Moreover, other contributing factors, such as oncogenic drivers like mutations occurring in the *KRAS* gene, also have an impact on the overall miRNA biogenesis and effector function, thereby contributing to broader miRNA dysregulation [80].

As a result, the dysregulation of miRNAs has attracted substantial interest from both academia and industry, standing as a pivotal research domain. This focus extends to comprehending disease biology and exploring their potential applications as diagnostic, prognostic, and predictive biomarkers [68]. Additionally, there is a growing interest in exploring miRNAs as potential drug targets or therapeutic agents [81].

miRNAs are widely recognized as potent genetic regulators that influence diverse biological and developmental processes, while also holding a pivotal role in the pathogenesis of various diseases. This potency stems from a single miRNA’s ability to regulate entire cellular pathways by interacting with numerous target genes [77].

Because of this, miRNAs have emerged as a novel class of therapeutic agents with the potential to restore disrupted cellular functions, particularly in various malignancies, including cancer. However, the very potency of miRNAs can be a double-edged sword. Their far-reaching effects, while beneficial, can also lead to off-target effects in non-targeted tissues, a concern documented in recent clinical trials [82,83,84]. Managing these off-target effects represents a significant challenge to be addressed. Take, for instance, MRX34, a miR-34a mimic encapsulated within a liposome-formulated nanoparticle (NOV40) that was evaluated in a first-in-human phase 1 study in patients with advanced solid tumors, including melanoma NSCLC, hepatocellular carcinoma, and renal carcinoma.

Despite MRX34 demonstrating significant efficacy, with three patients achieving prolonged confirmed partial responses and 14 patients maintaining stable disease (median duration, 136 days) [85], the clinical trial faced termination due to serious immune-mediated adverse events, leading to the deaths of four patients (NCT01829971) [82,83,84]. Nevertheless, the dose-dependent modulation of disease-associated target genes serves as evidence supporting the concept of miRNA-based cancer therapy.

This review discusses the dysregulation of miRNA expression in cancer and the potential of miRNAs as therapeutics. It also further discusses the primary challenges and strategies required to overcome obstacles and fully exploit the therapeutic potential of miRNAs.

## 2. miRNAs

Following the discovery of lin-4 as the first miRNA in 1993 in *Caenorhabditis elegans* [85,86], it became evident that miRNAs are widespread in the animal and plant kingdoms, some of which exhibit high levels of conservation across species [87,88,89].

miRNAs, short non-coding RNA molecules typically about 22 nucleotides long, are naturally encoded in the genomes of diverse species [87,88,89,90].

They play pivotal roles in regulating gene expression at both transcriptional [5,6,7] and post-transcriptional [8,9,10,11,91] levels of their target mRNAs [8,10], influencing mRNA stability and translation [92] across a wide array of biological processes [93], impacting activities such as cell differentiation, proliferation, angiogenesis, and apoptosis.

Additionally, miRNAs demonstrate distinct expression patterns in various tissues [12,13] and during different developmental stages [14,15,16].

There are currently estimated to be more than 2588 mature human miRNAs present in human cells [94], each with a unique temporal and tissue-dependent expression pattern. These miRNAs are estimated to control over 60% of human gene expression, showcasing their significant regulatory roles in diverse physiological processes. Because a single microRNA can regulate multiple genes, many miRNAs can contribute to the development of many human diseases when they become dysfunctional [2,8,18,20,21,22,23,24,25,26,28,30,31,32,33,34,35,37,66,95,96,97] including many types of cancer [39,41,42,43,44,47,50,51,53,55,56,57,58,59,60,61,62,63,64,65,66,67,69,98,99,100,101,102,103].

However, determining the precise relevance of individual miRNAs has been challenging, despite their evident significance as regulatory molecules [104]. Studies investigating miRNA functions through either suppression or overexpression of specific miRNAs have generated data that sometimes conflict with findings from loss-of-function models [104]. For example, studies in *Caenorhabditis elegans* involving systematic miRNA deletions suggest that fewer than 10% of the miRNAs are individually essential for normal development or viability [105] and this trend appears consistent in mice as well [96].

As illustrated in Figure 1, miRNAs are primarily transcribed from DNA sequences into primary miRNAs (pri-miRNAs), which undergo an initial processing step by Drosha within the nucleus to yield precursor miRNAs (pre-miRNAs) [8,68,106]. It is important to note that up to 40% of miRNA genes might be located within either the introns or exons of other genes [107]. After their transportation from the nucleus to the cytoplasm by exportin 5 (XPO5), pre-miRNAs undergo additional processing by endoribonuclease Dicer, leading to the formation of miRNA duplexes characterized by distinct 3′ overhangs of 2 nucleotides. Subsequently, these miRNA duplexes are loaded onto the Argonaute (AGO) protein, which retains one miRNA strand while discarding the other [10]. The AGO-miRNA complex, along with co-factors like GW182 (TNRC6A), forms the RNA-induced silencing complex (RISC) [91] responsible for cognate mRNA degradation and hence inhibition of translation through interaction with complementary mRNA sequences, typically located within the 3′-untranslated region (3′-UTR) of mRNAs (Figure 1) [108,109,110,111].

The interaction between miRNA and target mRNA typically takes place at the 5′ end of the miRNA, referred to as the ‘seed’ region. Yet, recent evidence points to a unique group of target mRNAs that bind the miRNA, not just through the seed but also via a complementary region at the 3′ end of miRNAs. This extended complementarity displaces the miRNA from Ago2, rendering it vulnerable to enzymatic degradation. This process is referred to as the target-directed miRNA degradation mechanism (TDMD) [112,113].

miRNAs are regarded as master regulators of the genome because of their capability to bind to and modify the expression of numerous protein-coding RNAs [114]. Because of this, a single miRNA can potentially regulate distinct mRNAs (anywhere from 10 to 100 protein-coding RNAs) due to their ability to bind to target mRNAs even when the pairing is not perfect [55,115]. As a result, a single miRNA can regulate a range of targets involved in similar cellular processes and pathways, thereby amplifying the cellular response potentially making miRNAs powerful therapeutics to restore perturbed cell functions seen in disease phenotypes. Conversely, a specific messenger RNA can become the target of many miRNAs, whether concurrently or in a context-dependent manner [116], leading to a collaborative repression effect [117,118]. Bioinformatic analyses indicate that a single miRNA can potentially bind to as many as 200 distinct gene targets with various functions, such as transcription factors, receptors, and more (https://bitesizebio.com/24926/mysterious-mirna-identifying-mirnas-and-their-targets/) (accessed on 20 December 2023).

## 3. miRNAs’ Role in Cancer

Cancer, a complex and heterogeneous disease, is characterized by a sequence of genetic and genomic abnormalities that promote tumorigenesis [119]. These alterations in the genome influence gene function, frequently resulting from genomic aberrations such as chromosomal translocations, amplifications, deletions, insertions, single-nucleotide mutations, or epigenetic modifications. These genetic and epigenetic alterations often result in the activation of oncogenes and the suppression of tumor suppressor genes [120]. In addition, miRNAs have been identified as additional genomic regulators that also play a crucial role in various aspects of organismal development, normal physiological processes, and the development of disease, including many types of cancers [68]. It has been shown that miRNAs play a pivotal in all the known processes involved in cancer, such as proliferation, survival, metastasis, and apoptosis [114]. Data suggest that dysregulation of miRNA function, either through its loss or gain, contributes to cancer development by either upregulating or silencing specific target genes. As a consequence, utilizing miRNAs either as miRNA mimics or antagomirs could present a potent therapeutic strategy to interfere with key molecular pathways associated with cancer as such miRNAs have the capacity to regulate all the recognized hallmarks of cancer, either acting as tumor suppressors or promoting oncogenic processes. Several of these cancer hallmarks influenced by miRNAs are discussed in detail in the literature [65,66].

It is widely accepted that alterations in miRNA genes and their expression are influenced by genetic deletions or amplifications, epigenetic methylation of miRNA gene locations, and modifications affecting pri-miRNA regulation by transcription factors as well as factors involved in miRNA biogenesis, often alter miRNA expression and function across various cancer types [66].

In addition, changes in the miRNA biogenesis process can also impact the availability of target mRNAs, including those associated with the development of cancer [121]. When miRNAs or the machinery involved in miRNA processing are altered or dysregulated this often leads to the loss of normal homeostatic state, leading to malignant transformation, including various types of cancer [51,52,56,65,66,67,122].

Due to their pivotal role in regulating the expression of numerous genes implicated in cellular responses to environmental stressors like hypoxia, oxidative stress, DNA damage, and nutrient deprivation, miRNAs can serve either as oncogenes (oncomirs) or tumor suppressors (onco-suppressor miRs). This is supported by recent findings that have identified miRNAs with oncogenic and tumor-suppressing roles in a range of neoplastic malignancies, and the dysregulation of miRNA expression is closely linked to the initiation, progression, and metastasis of cancer [43,45,103].

Moreover, dysregulated circulating miRNAs have demonstrated associations with disease origin, progression, treatment response, and patient survival [123,124]. For example, the distinctive tissue specificity of miRNAs [13], crucial for the maintenance of normal cells and tissues [40], renders them promising candidates for potential biomarkers in diagnosing cancers of unknown primary [125,126].

Furthermore, with the frequent genetic and epigenetic changes identified in particular miRNAs and the elements involved in miRNA biogenesis across diverse cancer types, oncogenic and tumor suppressor miRNAs have emerged as promising candidates as miRNA-based therapeutics.

## 4. RNA Therapeutics

As detailed in the literature [127,128], over the past few years, more than 50 siRNA-based drugs have progressed into phase I–III clinical trials. Of those, around 15 programs based on siRNA therapeutics are currently being investigated in phase I, II, and III trials for the treatment of different cancer types [128].

Two siRNA-based drugs, Patisiran and Givosiran (both developed by Alnylam Pharmaceuticals (Cambridge, MA, USA)), obtained approval from the Food and Drug Administration (FDA, Washington, DC, USA) in 2018 and 2019. Patisiran was approved for hereditary transthyretin-mediated amyloidosis, while Givosiran gained approval for acute hepatic porphyria [129,130].

The first example of an FDA-approved RNA-based drug, a siRNA-based therapy developed by Alnylam Pharmaceuticals, is Patisiran, sold under the brand name Onpattro™ for the treatment of polyneuropathy of hereditary transthyretin-mediated amyloidosis in adults. Based on the completion of a successful phase III APOLLO trial, Onpattro™ was approved by the US FDA in August 2018. Onpattro™ contains patisaran, which comprises a siRNA targeting transthyretin (TTR) mRNA conjugated with a lipid complex which leads to a decrease in TTR protein levels in the liver, thus resulting in a reduction in amyloid deposits. Patisiran targets and binds to a genetically conserved sequence found in the 3′UTR of both mutant and wild-type TTR mRNA [131]. Findings from the APOLLO trial, a placebo (77 patients)-controlled phase III trial which enrolled 225 patients showed that 51% of patients receiving Onpattro™ (148 patients, once every three weeks (0.3 mg/kg body weight)) experienced an enhanced quality of life (measured using the Norfolk Quality of Life Diabetic Neuropathy (QoL-DN)), as compared to only 10% of patients in the control group, which received a placebo drug [131,132].

During the COVID-19 pandemic, mRNA technology became instrumental, notably in the development of highly effective mRNA vaccines. These vaccines have played a crucial role in controlling the transmission of severe acute respiratory syndrome coronavirus 2 (SARS-CoV-2).

The groundbreaking science behind mRNA vaccines earned Katalin Karikó and Drew Weissman the prestigious 2023 Nobel Prize in Physiology or Medicine for their pioneering work on nucleoside base modifications, enabling the development of these impactful COVID-19 vaccines.

The evolution of cap analogs has vastly improved mRNA translation, while advancements in purification, packaging, and delivery methods have revolutionized medicine. Visionaries like Katalin Karikó, Drew Weissman, Edward Darzynkiewicz, Robert Rhodes, Ugur Sahin, and Ozlem Tureci made pivotal early contributions to mRNA research, deserving recognition for their pioneering efforts. This narrative around mRNA charts a remarkable journey showcasing breakthroughs in a field holding immense promise for the future of medicine.

Consequently, the success of mRNA vaccines has paved the way for integrating mRNA-based technology into personalized neoantigen vaccines, seamlessly incorporating them into the standard oncological workflow [133,134]. These mRNA-based vaccines can be tailored and manufactured as individualized vaccines with multiple neoantigens [135], and can effectively stimulate antigen-presenting cells [136,137,138,139] and be delivered using clinical-stage delivery formulations [140]. The studies and insights from the mRNA-based COVID-19 vaccines highlight the promise of RNA therapeutics as an innovative class of treatments.

However, the effectiveness of miRNA and other nucleic acid-based therapies hinges on a reliable delivery method with minimal adverse events and drug- or treatment-related toxicity.

Delivering miRNA therapeutics to cells poses challenges, requiring precise targeting of diseased cells while sparing healthy ones. In contrast to mRNA COVID-19 vaccines, which are taken up by scavenging immune cells such as dendritic cells and other professional antigen-presenting cells, thus inducing a specific immune response through the processing and expression of the translated mRNA molecules [141], miRNA therapeutics must effectively bypass immune recognition to reach their target cells without triggering an immune response.

## 5. miRNA Therapeutics

The discovery of the link between miRNAs and human diseases in 2002 sparked a strong interest in their potential as a new class of therapies. Consequently, interdisciplinary fields encompassing biology, chemistry, and medical science have made significant investments in the development of miRNA-based therapies.

As illustrated in Figure 2 and Table 1 and Table 2, and discussed in the literature in detail [77,121,127], there are only a few miRNA therapeutics that have entered clinical trials with none of them entering phase III or being approved by the FDA and several of them having been terminated due to toxicity. Despite considerable advancements in preclinical research, the field of miRNA-based diagnostic [68] and therapeutic applications is still in its early stages. Only a few of these miRNA-based therapies have progressed to clinical development. Given this situation, several efforts in the biotechnology and pharmaceutical industry have integrated miRNAs into their development pipelines focusing on the development of two categories of miRNA drugs, miRNA mimics and inhibitors (antagomirs or antimirs) (Figure 2, Table 1 and Table 3) [77,142]. As a result, the number of miRNA-based therapeutics being tested in clinical trials for various genetic, metabolic, and oncological conditions is continually increasing conditions [143,144,145].

miRNA therapeutics are a type of RNA-based therapeutic that targets and modulates the activity of specific endogenous miRNAs in the body. Because miRNAs play a critical role in regulating gene expression under normal health and pathological conditions, by targeting and manipulating specific miRNAs, miRNA therapeutics aim to treat various diseases by restoring (miRNA mimics) or correcting miRNA expression patterns (antagomir).

As discussed in the literature [175], the journey of discovering and developing miRNA therapeutics begins with target identification and miRNA discovery through the analysis of patient samples. This process involves in silico miRNA validation by pairing genomic databases with biological validation and in silico miRNA target identification. Further steps include validating miRNA relevance to a specific disease using tissue culture and in vivo models.

To improve the stability of miRNAs, various chemical modifications are applied to miRNAs, such as 2′-O-methyl, 2′-F, LNA, PS, and PNA, among others.

Additionally, conjugation with different biomolecules and optimizing delivery systems, such as ligand-targeted lipid nanoparticles are used to achieve precise delivery of RNA molecules to targeted tissues or cells for in vivo applications.

The subsequent stages encompass ADME and preclinical PK and PD studies, efficacy, and toxicity testing in relevant in vivo models, GMP manufacturing, scale-up studies, IND filing, and thorough evaluation of dosage, safety, and efficacy in Phase I-III human clinical trials. This intricate and resource-intensive process culminates in safety and efficacy assessments leading to FDA approval.

miRNA-based therapeutic programs for cancer are predominantly conducted by a few biopharmaceutical companies, including Santaris Pharma (Hersholm, Denmark), Roche Pharmaceuticals (Singapore), Regulus therapeutics (San Diego, CA, USA), Mirna Therapeutics Inc. (Carlsbad, CA, USA), miRagen Therapeutics (Boulder, CO, USA), and EnGeneIC (Sydney, Australia).

Immune evasion and chemotherapy resistance is a challenge in cancer therapy and this resistance can be mediated by numerous factors including miRNAs induced by tumor microenvironment stimuli, like hypoxia or cell–cell communication [176]. For example, hypoxia has been shown to influence miRNA expression in cancer and stromal cells in the tumor microenvironment (TME) via downregulation of factors involved in miRNA biogenesis or modulation of transcription factors that control miRNA expression. Accordingly, many hypoxia-regulated miRNAs and their role in tumor progression have been reported.

These hypoxia-regulated miRNAs, such as miR-26a, miR-181b, miR-210, miR-301a, miR-424, and miR-519c, have been associated with responses to chemotherapy or radiotherapy across various cancers [177,178,179,180].

Therefore, targeting these miRNAs therapeutically presents a potential strategy to resensitize hypoxic tumors to chemotherapy and other treatments. For instance, within the hypoxic microenvironment of pancreatic cancer, HIF-1α induces resistance to gemcitabine.

A recent study has demonstrated that administering miR-519c, which exhibits decreased levels in pancreatic cancer, inhibited HIF1-α within gemcitabine-resistant pancreatic cancer cells under hypoxic conditions [181].

Additionally, a redox-sensitive nanoplatform was employed to simultaneously administer miR-519c and gemcitabine [181]. This approach effectively suppressed the expression of HIF-1α and genes responsible for glucose uptake and cancer cell metabolism, thus inhibiting the growth of orthotopic desmoplastic pancreatic cancer in NSG mice and reversing the chemotherapy resistance induced by hypoxia [181].

Likewise, the tumor suppressor miR-34a has been demonstrated to decrease the expression of over 30 oncogenes spanning various oncogenic pathways and genes involved in tumor immune evasion; however, its expression is often lost or reduced in numerous malignancies [82].

However, if miRNAs are to be used for the treatment of a cancer, miRNAs must be delivered to the target tissue, not trigger an immune response, and be economically feasible so that wide-spread adoption of these nano therapies can be realized. While some clinical progress has been achieved, several trials have faced termination, primarily due to serious adverse events and drug induced toxicity. These setbacks underscore the existing challenges that must be addressed before miRNA-based therapies can find broad clinical application.

### 5.1. Examples of miRNA Therapeutics in Clinical Trials

As illustrated in Figure 2, and Table 1 and Table 2, and discussed in detail in the literature [145,182], there are several miRNA-based therapeutics being tested both in preclinical studies (Table 1) or in human clinical trials (Table 2).

Miravirsen: Miravirsen (SPC3649), the pioneering miRNA therapeutic to enter clinical trials, is a 15-mer LNA-PS-modified ASO antagomir targeting miR-122. Developed by Santaris Pharma and Roche Pharmaceuticals, it is aimed at targeting hepatitis C virus (HCV) infections.

miR-122 has been shown to play a role in HCV replication [165]. Phase II clinical trials were conducted to evaluate the safety and antiviral efficacy of Miravirsen in patients with chronic HCV infection.

Miravirsen demonstrated significant efficacy in reducing viremia among HCV patients [168,169,183], leading to several phase II clinical trials (NCT01200420, NCT01872936, NCT02031133, NCT02508090). However, the trial was terminated due to severe side effects [145,184].

RG-012: RG012 is an anti-miR-21 therapy developed by Regulus Therapeutics for the management of Alport syndrome (fibrotic kidney disease). miR-21 has been shown to be upregulated in Alport syndrome. Preclinical studies have indicated that administering an anti-miR-21 significantly mitigated kidney failure by reducing the progression rate of renal fibrosis. Based on these robust preclinical data, RG-012 has been granted orphan drug status in the US and Europe. However, some sequence-independent side effects have been reported in relation to phosphorothioate-modified oligonucleotides [185].

RG-101: RG-101, an antagomir of miR-122 developed by Regulus Therapeutics, is an N-acetyl-d-galactosamine(GalNAc)-conjugated synthetic RNA oligonucleotide that targets and inhibits miR-122, which is involved in HCV replication for patients with HCV. In addition to its essential role in HCV replication, miR-122, a liver-specific miRNA, has relevant functions in liver metabolism [186]. miR-122 also acts as an essential host factor for HCV replication.

Clinical trials were conducted to evaluate RG-101’s safety and efficacy as a potential treatment for chronic HCV infection. Although RG-101 showed considerable efficacy and a significant decrease in viral loads among chronic HCV patients [169], the trial was terminated due to some serious adverse events of severe hyperbilirubinemia [169].

MRG-201: MRG-201 is a synthetic RNA oligonucleotide that targets and activates miR-29, which has been shown to inhibit fibrosis. Clinical trials have been conducted to assess MRG-201’s safety and efficacy in treating fibrotic disorders such as hypertrophic scars and idiopathic pulmonary fibrosis.

MRX34: MRX34, developed by miRNA Therapeutics Inc. is a synthetic miRNA mimic designed to mimic the activity of a tumor suppressor miR-34a encapsulated into a liposome-formulated nanoparticle (NOV40) for the treatment of advanced solid tumors including melanoma, NSCLC, hepatocellular carcinoma, and renal carcinoma.

miR-34a, a natural tumor-suppressor miRNA, is often expressed at diminished levels in various tumor types. MRX34, recognized as a first-in-class miRNA mimic therapeutic, is designed for treating several cancers including non-small cell lung cancer, hepatocellular carcinoma, colon cancer, ovarian cancer, cervical cancer, and more. This formulation underwent evaluation in a phase I clinical trial [82]. While MRX34 showed promising activity, the trial concluded with only three patients achieving sustained confirmed partial responses and 14 patients maintaining stable disease (median duration of 136 days) [182]. Unfortunately, due to severe immune-mediated adverse events leading to the deaths of four patients, the trial was terminated (NCT01829971) [82,83,84]. After this, MiRNA Therapeutics ceased operations in 2017 and agreed to merge with Synlogic Inc. (Cambridge, MA, USA).

Cobomarsen (MRG-106): MRG-106 (Cobomarsen), an LNA-based antagomir of miR-155 was developed by Miragen Therapeutics (Viridian Therapeutics Inc., Waltham, MA, USA) that aimed to inhibit the activity of miR-155 [171,187] in several lymphoma subtypes, as well as in diffuse large B-cell lymphoma [101] where miR-155 is upregulated. Phase II clinical trials are being conducted to assess its effectiveness in treating certain cancers and immune disorders including cutaneous T-cell lymphoma (CTCL), chronic lymphocytic leukemia, diffuse large B-cell lymphoma, and mycosis fungoides (NCT03837457), and adult T-cell leukemia/lymphoma (NCT02580552, NCT03713320). While the phase I trial was completed, two of the phase II studies were terminated. The study was prematurely terminated due to business reasons, not due to safety concerns or lack of efficacy (https://classic.clinicaltrials.gov/ct2/show/NCT03713320) (accessed on 20 December 2023).

MRG-107: MRG-107, an antagomir of miR-155 developed by Miragen Therapeutics (Viridian Therapeutics Inc.), aimed to inhibit the activity of miR-155. miR-155 plays significant roles in immune mechanisms and inflammation processes within amyotrophic lateral sclerosis (ALS), where its levels are elevated in the spinal cords of ALS patients. In preclinical models of ALS, inhibition of miR-155 has reduced ALS symptoms and extended survival [145].

MRG-110: MRG-110 is a synthetic antagomir of miRNA-92a developed by MiRagen Therapeutics in collaboration with Servier to treat ischemic conditions such as heart failure [154].

MRG-110 was designed to inhibit miR-92a and stimulate the growth of new blood vessels. MRG-110 is being investigated to assess its potential to improve wound healing by enhancing blood circulation within the wound site. A phase I human clinical trial to test the safety, tolerability, pharmacokinetics, and pharmacodynamics of MRG-110 following intradermal injection in healthy volunteers was recently completed (NCT03603431).

Remlarsen (MRG-201): Remlarsen (MRG-201), an LNA RNA mimic of miR-29 developed by MiRagen Therapeutics, is intended for keloid disorder. Remlarsen reduces the expression of collagen and other proteins involved in scar formation, exhibiting inhibitory effects on fibrosis [161].

miR-29 family members are typically downregulated in fibrotic diseases [188]. A phase II clinical trial (NCT03601052) is currently underway to evaluate Remlarsen’s safety and efficacy, such as if it can limit the formation of fibrous scar tissue in certain fibrotic disorders like hypertrophic scars and idiopathic pulmonary fibrosis. This evaluation involves administering Remlarsen through intradermal injection at the location of an excisional wound [161].

TargomiRs: The MesomiR 1 trial (NCT02369198) tested the safety and efficacy of miR-15/16 encapsulated in nonliving bacterial minicells (nanoparticles, referred to as TangomiRs) in patients with recurrent malignant pleural mesothelioma (MPM).

TangomiRs were developed by EnGeneIC to deliver miR16 mimics encapsulated in TargomiRs composed of bacterial minicells with an anti-EGFR bispecific antibody to target EGFR-expressing cancer cells. TargomiRs were evaluated as 2nd- or 3rd-line treatment for patients with recurrent malignant MPM and non-small-cell lung cancer (NSCLC) (NCT02369198) [157]. miR-15/16 are implicated as tumor suppressors in MPM.

Although variable response rates were observed, with 5% of the patients exhibiting a partial response, 68% exhibiting stable disease, and 27% displaying progressive disease following low-dose systemic administration of TargomiRs, dose-dependent toxicities emerged, such as anaphylaxis, inflammation, and cardiac events [127,157].

MGN-1374: MGN-1374, an 8-mer LNA ASO developed by miRagen Therapeutics, is designed to specifically target the seed region of the miR-15 family and is currently in the preclinical phase for the control of postmyocardial infarction remodeling.

RGLS4326: RGLS4326 is a chemically modified, single-stranded, 9-mer ASO that possesses full complementarity to the seed sequence of miR-17. RGLS4326 is specifically developed to inhibit the pathological functions of the miR-17 family in autosomal dominant polycystic kidney disease (ADPKD) [189], one of the most frequent monogenic disorders, caused by mutations in the PKD1 or PKD2 gene and for which therapeutic options are limited. A phase I clinical trial of RGLS4326 was recently completed (NCT04536688).

Additionally, Table 1 highlights several miRNA-based drugs currently under preclinical investigation, targeting various diseases such as peripheral arterial disease, chronic heart failure, and amyotrophic lateral sclerosis (ALS), among others.

Moreover, miRNA therapeutics in combination with chemotherapeutic agents have also been explored to overcome cancer therapy resistance [142]. Studies indicate that combining therapeutic miRNAs with chemotherapy can decrease the required drug doses for cancer treatment [190,191]. For example, miR-3622b-5p, when paired with cisplatin, not only enhances apoptosis but also sensitizes ovarian tumor organoids to cisplatin [192], suggesting the potential of miRNAs in combination with chemotherapy to address cancer treatment and counteract drug resistance.

### 5.2. Small-Molecule Modulators of miRNA Expression

Because altered levels of miRNA expression are associated with many cancers, restoring the function of tumor suppressor miRNAs by overexpressing or introducing of miRNA mimics to restore to their relatively normal physiological levels or function or by inhibiting overexpressed oncogenic miRNAs by miRNA inhibitors (antagomirs) or miRNA sponges represents two major strategies for miRNA therapeutics in cancer [121,193] (Figure 2).

The function of repressed miRNAs can also be restored to their relatively normal physiological levels by using some small molecules that can transcriptionally activate the expression of miRNA genes leading to the expression of endogenous miRNAs and restoring the expression of tumor-suppressive miRNAs. Conversely, overexpressed oncogenic miRNAs can also be suppressed by small-molecule inhibitors.

As illustrated in Figure 3 and discussed in the literature [121], because nucleic acid-based therapeutics have poor cell-permeability for drug delivery, in recent years small-molecule drugs in the regulation of miRNA expression have been explored since they can cross the cell membrane by free diffusion and can modulate the expression of miRNAs and also traditional drug development can be applied for the development of novel miRNA inhibitors (or activators) [194]. Various small-molecule inhibitors of various miRNAs with various chemical structures and different mechanisms of action that interferes with entire miRNA biogenesis process, including processing, maturation, and function, have been described in the literature [121] (Figure 3). For example, as discussed in the literature [121], reduced expression of tumor suppressor miRNAs can be reactivated to their normal physiological levels by some small molecule compounds, such as hypomethylating agents [195]. Two hypomethylating agents, decitabine or 5-azacytidine used for the treatment of myelodysplastic syndrome have been shown to upregulate the expression of many miRNAs [105].

Similarly, enoxacin was shown to activate the expression of several miRNAs in vitro [196] and to suppress tumor growth by promoting miR 24 expression in vivo [196] suggesting that small-molecule compounds can potentially restore miRNA expression and function to a more physiological setting.

For example, miR-21, a tumor-associated miRNA (oncomir) has been shown to be upregulated in many cancers (e.g., breast cancer, colon cancer, ovarian cancer, pancreatic cancer, thyroid cancer, and others) and to exhibit high expression in cancer which is closely associated with tumorigenesis [197,198,199,200]. Trypaflavine (TPF), a small-molecule inhibitor of miR-21 [201], has been shown to inhibit RISC formation by disrupting the interaction between miR-21 and the AGO2 protein, consequently resulting in the suppression of miR-21 expression levels. Likewise, streptomycin has been shown to inhibit the cleavage of pre-miR-21 by Dicer [202]. Similarly, small-molecule compound AC1MMYR2 has been shown to blockade the cleavage of pre-miR-21 to generate mature miR-21 [203]. Furthermore, diazobenzene, azobenzene, and estradiol have been shown to inhibit the transcription of the miR-21 [204,205], while, polylysine has been shown to inhibit the maturation process of pre-miR-21 by inhibiting Dicer [201]. Similarly, arylamide derivatives have been shown to inhibit miR-21 maturation [206]. 4-benzoylamino-N-(prop-2-yn-1-yl)benzamides has been shown to increase the expression of PDCD4, the functional target of miR-21 [207].

As for other miRNAs, kanamycin A has been shown to inhibit let-7 expression by attaching to pre-let-7 and disrupting the Dicer function [208].

Additionally, small-molecule inhibitors such as NSC 158959 and NSC 5476 have been implicated in potentially inhibiting miR-122 gene transcription [209]. Crucially, mir-122 represents a liver-specific miRNA, constituting approximately 72% of the total miRNA content within the adult liver. It stands out as one of the initial miRNAs displaying high levels of tissue-specific expression [210]. miR-122 plays a pivotal role in regulating cholesterol and fatty acid metabolism within the adult liver, positioning it as a promising therapeutic target for metabolic diseases [211]. Additionally, miR-122 is instrumental in the progression of diverse liver conditions, encompassing acute and chronic liver injury, liver tumors, hepatitis C virus (HCV) infection, liver cirrhosis, and alcoholic hepatitis [212]. Furthermore, research has demonstrated that miR-122 plays a crucial part in the onset and progression of acute and chronic liver injuries, liver tumors, hepatitis C virus (HCV) infections, liver cirrhosis, and alcoholic hepatitis [212].

As for miR-96, benzimidazole has been shown to selectively inhibit biogenesis of miR-96, upregulating a protein target (FOXO1) and inducing apoptosis in cancer cells [213].

miR-1, found in high levels within skeletal muscle, contributes to regulating the creation of the skeletal muscle cells and the overall development of muscles and is associated with heart development [214,215]. A number of small-molecule inhibitors targeting miR-1 have been discovered via photocycloadducts of acetylenes combined with a basic structure of 2-methoxy-1,4-naphthalenequinone through a photocyclization reaction [216]. For instance, 2-methoxy-1,4-naphthalenequinone was identified for its selective inhibitory impact on miR-1 expression in cells. However, the precise mechanism by which 2-methoxy-1,4-naphthalenequinone exerts its inhibitory function on miR-1 expression remains to be fully understood.

Furthermore, arsenic has been demonstrated to inhibit the transcription of the miR-27a gene [217]. On the other hand, trioxide, neomycin, amikacin, and tobramycin were found to exert their inhibitory action by targeting Dicer, consequently impeding the maturation process of miR-27a [218].

Likewise, 5″-azido-neomycin B has been shown to inhibit miR-525 by binding to the processing site of Drosha to disrupt the production of pre-miR-525 [219].

N-substituted oligoglycines, a particular peptoid ligand with the apical loop of pri-miR-21, have been shown to suppress the processing of pri-miR-21 by Drosha by binding to pri-miR-21 [220].

## 6. Advances in the Delivery of miRNA Therapeutics

While a handful of phase 1 and 2 clinical trials have explored miRNA-based therapeutics, there are currently no miRNA-based therapeutics undergoing phase III human clinical trials. This is partly attributed to challenges associated with precisely delivering miRNAs to specific cell types, tissues, and organs.

While several approaches, such as antibodies, nanoparticles, or ligands, have been documented to enhance the effectiveness of miRNAs and decrease off-target effects (like immunotoxicity [221]) when directing miRNAs to specific cells of interest, limitations and challenges persist in the field of miRNA therapeutics.

As illustrated in Figure 4 and discussed in detail in the literature [77,84,222,223], there are various strategies being explored as mechanisms to deliver miRNA therapeutics (mimics and antagomirs) to the indented tissue and to improve pharmacokinetic mechanisms, and avoid off-target effects.

These methods include both vector and non-vector approaches, ranging from lipid-based nanoparticles, polymeric vectors, dendrimer-based vectors, cell-derived membrane vesicles, 3D scaffold-based delivery systems, to various biodegradable and biocompatible nanoparticles derived from polymers and metals [77].

Other strategies for the delivery of RNA-based therapeutics, as discussed in the literature [225], include adeno-associated virus, lentivirus, bacterial nanocells [226], bacteriophages, cationic lipid-based liposomes (including monovalent and multivalent lipids), natural polymer-based nanoparticles, polymer based nanoparticles conjugated with polyethylene glycol (PEG), extracellular vesicles (EVs) or exosomes, nanocomplex-forming functionalized metals such as gold nanoparticles, and carbon nanotubes, polymeric micelles, and mesoporous silica nanoparticles [145], and many others that are engineered to contain biomolecule conjugates for improved stability and pharmacokinetics and target delivery to the intended cell or tissue type [84,223,225,227,228]. A recent study showcased the increased antitumor effectiveness of STING agonists through the covalent attachment of cyclic dinucleotides (CDN) to polymer nanoparticle (poly(β-amino ester) formulation for intravenous delivery [229].

Non-pathogenic recombinant viral vectors, such as retroviruses and lentiviruses (which pose a genomic integration risk), adenoviruses, and adeno-associated viruses (which remain transiently stable in an episomal form within the host cell’s nucleus) [230,231], are under investigation for their capacity to encode the desired RNA transgene. These vectors are being explored for intracellular delivery of miRNA-based therapeutics, making them a significant area of interest [232].

A phase II trial is currently testing an adeno-associated viral vector for the delivery of the miRNA drug AMT-130 for the treatment of Huntington’s disease (ClinicalTrials.gov identifier NCT04120493) [146,147,148]. Despite their potential for the delivery and expression of miRNAs, there are various side effects with the use of viral vectors such as immunogenicity and transgene-related immune responses [233].

The packaging of the negatively charged nucleic acids in liposome nanoparticles masks their negative charge and also protects against serum nuclease degradation [228,234]. Delivery of miRNAs using liposome nanoparticles has already been applied in several clinical studies, such as MRX34 (NCT01829971, NCT02862145) [82,83].

Likewise, bacterial minicells loaded with miRNAs were employed to deliver miR-16 mimics during a phase 1 trial involving patients with recurrent malignant pleural mesothelioma (MesomiR 1, NCT02369198) [156,157]. However, the study also reported several side effects including dose-limiting toxicities, decreased lymphocyte counts, or cardiac events [157].

Extracellular vesicles (EVs), including exosomes, are under exploration as potential drug delivery systems, capable of delivering specific genetic cargo for cellular transfer within the body [235]. For example, EVs derived from mesenchymal stromal cells obtained from human adipose tissue were modified to carry miR-125b. This modification resulted in the inhibition of human hepatocarcinoma cell proliferation [236].

In addition, different modalities of drug delivery systems have been explored for the delivery of miRNA-based drugs such as core–shell magnetic nanoparticles, quantum dot nanocrystals, polymeric micelles, and mesoporous silica nanoparticles are among the other examples of nanocarriers as drug-delivery systems to improve the therapeutic effectiveness and specificity, and tissue targeting of miRNA and other nucleic acid therapeutics [145].

An encouraging strategy involves the covalent conjugation of miRNAs, along with other nucleic acid-based drugs and biomolecules, to lipids, peptides, or sugars. These compounds function through receptor-mediated endocytosis mechanisms [228].

Likewise, a lipophilic cholesterol conjugate was employed to deliver an miR-29-based mimic (remlarsen/MRG-201) to human skin fibroblasts, irrespective of cell type via skin injection in a phase II trial for keloid disorder. The aim was to suppress the expression of extracellular matrix and fibroplasia within the skin (NCT02603224, NCT03601052) [161].

Another approach involves the coupling of aptamer conjugates to specific miRNA therapeutics using a straightforward sticky-end annealing method [237]. This method serves as a strategy for delivering miRNAs to targeted cell types.

Aptamers are single-stranded nucleic acids that are developed as high-affinity ligands specific to cell surface receptors to facilitate the delivery of therapeutic cargo including miRNAs through receptor-mediated transport [228,237]. In ongoing preclinical investigations, researchers are currently investigating aptamer-linked miRNAs, such as the Aptamer-miR-34c conjugate (known as GL21.T-miR-34c) in non-small-cell lung cancer cells [238].

Additional efforts have been made to enhance the serum stability, pharmacokinetics, and tissue specificity of miRNA mimics, miRNA inhibitors, and other nucleic acid therapeutics by incorporating various chemical modifications to miRNA and nucleic acid therapeutics or attaching various biomolecule conjugates to these therapeutic miRNAs to facilitate receptor-mediated uptake such as N-acetylgalactosamine (GalNAc), 2′-O-methyl nucleotide, phosphorothioate, cholesterol, locked nucleic acid (LNA), and aptamer moieties are also shown as examples [77,84,153,223,239].

For example, biomolecule conjugates, such as N-acetylgalactosamine (GalNAc), have been investigated in clinical trials. GalNAc facilitates the targeted delivery of nucleic acid therapeutics through endocytosis by activating liver cell-specific asialoglycoprotein receptors [240,241]. GalNAc linked to a miR-122 inhibitor (RG-101) and a miR-103/107 inhibitor (RG-125/AZD4076) are in clinical trials for chronic HCV [EU Clinical Trials Register (clinicaltrialsregister.eu) EudraCT numbers 2015-001535-21, 2015-004702-42, 2016-002069-77] and steatohepatitis (NCT02612662, NCT02826525), respectively [152,153,241]. However, due to reported side effects such as cases of jaundice, the clinical trial for RG-101 was halted, and investigations into the cause of these effects are ongoing [150,169,170].

Recent preclinical investigations have also explored other examples of GalNAc-conjugated LNA, anti-miR-122 antisense oligonucleotides, or nano-carrier vehicles in combination with cell type-specific biomolecule conjugates or miR-155 inhibitors by gold nanoparticles formulated with antagomir and AS1411 aptamer [242,243].

In addition, the 3D matrices for delivering nucleic acid-based therapeutics and conventional drugs are currently undergoing optimization with diverse design features. This encompasses various application routes, such as edible or injectable carriers [77,244,245,246].

One potential method of administering miRNAs is orally [75]. Data demonstrate that miRNAs are commonly associated with EVs, lipoproteins, or lipid derivatives, and RNA-binding proteins. These associations, along with the use of nanoparticles, shield miRNAs from the harsh conditions in the gastrointestinal tract. This includes protection against salivary and pancreatic RNases, the stomach’s acidic pH, digestive enzymes, peristaltic activity, and microbial enzymes. This safeguarding mechanism is thought to aid in the absorption of miRNAs from the digestive tract [75]. However, there is ongoing debate surrounding the absorption, stability, and physiological impact of these edible or food-derived miRNAs.

## 7. Progress in Chemical Modifications of miRNAs for Improved Stability and Cellular Uptake

As depicted in Figure 4 and discussed in the literature [77], the combination of various chemical modifications, conjugation of various types of biomolecules, or the utilization of carriers increases stability and improves precise delivery of RNA molecules to targeted tissue or cells (i.e., disease site).

Moreover, a range of chemical alterations to nucleobases, ribose sugars, or the phosphate backbone can conceal the negative charge of miRNAs and other nucleic acids. This modification enhances their adherence to the cell surface, subsequently aiding in cellular uptake [190,216] and bolstering their stability as well.

In addition, a range of chemical modifications to nucleobases, ribose sugar, or the phosphate backbone can mask the negative charge of the miRNAs and other nucleic acids. This modification, the adhesion of miRNAs to the cell surface, subsequently aids in cellular uptake [228,247] and bolsters their stability as well.

One frequently utilized form of nucleic acid modification is locked nucleic acid (LNA) bases [247], which involves the insertion of methylene bridges. These methylene bridges decrease the flexibility of the ribose ring, resulting in a locked conformation of the modified nucleotides [248,249]. LNA-modified RNA-based therapeutics are more resistant to ribonucleases and exhibit improved cellular uptake, primarily through an endocytosis mechanism that remains not fully understood [250]. Furthermore, the locked conformation significantly boosts the ability of LNA-based RNA therapeutics to form stable duplexes, enabling them to bind and effectively inhibit the function of the targeted miRNA. Consequently, LNA-modified RNAs are frequently employed in single-stranded antagomirs, such as antisense oligonucleotides (ASOs) [247,250]. Due to these benefits, LNA-modified oligonucleotides have emerged as one of the primary approaches for inhibitory therapeutics targeting both miRNA and mRNA. This alteration enhances the stability of the oligonucleotide and facilitates its uptake into endosomes.

As discussed in the literature [251], the cellular uptake of LNA-modified oligonucleotides and other naked oligonucleotides is primarily mediated through endocytosis. Multiple pathways and cellular proteins, including clathrin, dynamin, and caveolar-dependent endocytosis are involved in this process. Notably, clathrin-independent endocytosis is also a mechanism, and studies have demonstrated that the antisense activity in hepatocytes can be hindered by the adaptor protein AP2M1, but not by clathrin and caveolin.

Phosphorothioate modifications involve the introduction of a sulfur atom into the oligonucleotide’s phosphodiester backbone. Phosphorothioate modifications enhances the stability of the oligonucleotide and facilitates its uptake into endosomes, particularly via stabilin receptors located on cell surfaces (such as those of kidney cells) [252,253]. This approach was applied for the specific delivery of synthetic miR-21–anti-miR (RG-012/lademirsen/SAR339375) into the kidney in a clinical study of Alport syndrome (NCT03373786, NCT02855268) [149]. However, specific sequence-independent effects have been reported in connection with phosphorothioate-modified oligonucleotides [185].

Another type of modification is a pH low-insertion peptide (pHLIP)-modified antimir that has been shown to inhibit the oncomir miR-155 in lymphoma [175,254]. pHLIP is a small peptide that forms a transmembrane α-helix under acidic conditions such as in a tumor microenvironment [255].

Likewise, a recent study has introduced a fully modified miR-34a (FM-miR-34a) designed to address concerns regarding miR-34a stability, non-specific delivery, and associated delivery-related toxicity [69]. FM-miR-34a exhibited potent suppression of proliferation and invasion, resulting in a prolonged inhibition of its target genes for more than 120 h after the in vivo administration of FM-miR-34a conjugated to folate (FM-FolamiR-34a).

This treatment resulted in the inhibition of tumor growth, leading to complete cures in some mice [69]. There were no notable changes observed in the body weight of mice during the study, suggesting that FM-FolamiR-34 is safe. These findings hold promise for reinvigorating miR-34a as a potent anti-cancer treatment, offering a compelling basis for clinical trials.

## 8. Progress in Predicting and Validating miRNA Targets

As discussed recently in detail by Diener et al. [77], the significant challenges for miRNA-based therapeutic approaches stem from their pleiotropic effects in regulatory networks [256]. Because of this, a comprehensive functional characterization of individual candidate miRNAs is essential before considering them for therapeutic purposes. This characterization involves overcoming several issues that must be addressed such as confirming the authenticity of a miRNA as a true miRNA, enhancing miRNA target prediction algorithms, and implementing experimental strategies that enable efficient validation of a large number of miRNA targets.

As discussed in the literature [182], the mechanistic functions of candidate miRNAs can be evaluated through bioinformatic analysis and/or in vitro experiments before progressing to testing in preclinical animal models. In addition, multiple databases and algorithms have been developed and are available for predicting their targets associated with each miRNA [108,257]. To enhance the predictive accuracy of miRNA target prediction, it is common practice to employ multiple distinct algorithms simultaneously to predict miRNA-binding sites in protein-coding genes and relevant biological pathways and networks. As discussed in detail in the literature [77], several computational tools are available for predicting miRNA–gene interactions [258]. Initially, target predictors for individual miRNAs, such as TargetScan [259] or miRanda [260] were developed. An example algorithms is TargetScan [108], which predicts miRNA targets based on seed regions that are essential for mRNA binding. TargetScan encompasses nearly all miRNA sequences documented in miRBase to date. These tools suggest target genes, forming the foundation for modeling the impact of miRNAs on genes and aiding in the selection of optimal targets for validation. However, they may lack specificity or sensitivity in predicting actual targets. Subsequently, more comprehensive analysis tools were created to model the mutual influence of miRNAs on genes and pathways, utilizing predicted and validated targets along with expression data and/or sequencing information [261]. For instance, miRTarVis [262] generates coexpression networks of paired miRNA and mRNA data, MIENTURNET [263] constructs interaction networks with enrichment analysis, miRViz [264] visualizes networks across multiple species, miRNet [265] supports statistical analysis and aids in the exploration of miRNA–target interaction networks, miTALOS [266] assesses miRNA function in a tissue-specific manner, and miRTrail [267] integrates miRNA and gene expression data for analysis. To enhance the specificity of target predictions beforehand, pathway databases can be incorporated. Because miRNA target genes can orchestrate entire pathways, integrating information on which putative target genes are enriched in functional biochemical networks significantly improves the validation rates of target predictors [268]. Lastly, tools for the systematic analysis of miRNAs concerning target genes or vice versa, for incorporating both validated and predicted targets, target pathways, and other information have been developed. An example is miRTargetLink2 [269], which utilizes miRNA gene associations from databases like miRPathDB2 [270] or mirDIP [271]. To enhance the specificity of functional effects of target predictions, these tools often utilize internet resources such as existing application programming interfaces to web services and online tools for in silico pathway analysis of miRNAs and target genes present in the interaction graph [272]. A recent publication offers an overview of miRNA target analysis tools [258].

Likewise, advances in high-throughput screens and bioinformatic tools for target prediction have significantly facilitated the study of miRNAs and prediction of their putative targets and biological pathways. For example, bioinformatics tools such as KEGG and ingenuity pathway analysis not only predict potential biological pathways but, in some instances, also identify disease states that may be influenced by miRNAs.

Furthermore, mechanistic modeling that simulates miRNA-mediated pathways has been investigated in recent works. Analyzing the function of miRNAs in gene regulatory networks can be aided by mathematical modeling. A comprehensive overview of the most recent developments, employing various mathematical modeling approaches to offer quantitative insights into the function of miRNAs in the regulation of gene expression, is discussed in recent reviews [273,274].

Additionally, several computational tools are available to calculate the free energy between RNA sequences of interest [275]. For instance, a lower free energy, usually around −20 or lower, indicates a stronger binding [275]. Thus, the integration of clinical research databases with miRNA bioinformatics platforms could enhance the identification and evaluation of potential therapeutic candidates.

As for the preclinical models, various human cell lines and induced-pluripotent stem (IPS) cells have been used to investigate the mechanisms, toxicity, and potential therapeutic efficacy of miRNA candidates as well as epigenetic manipulation of target transcripts [182]. For example, use of IPS cells enable modulation of biological pathways across various stem cell lineages derived from easily accessible skin tissue sources [276,277]. Furthermore, the availability of various animal models through academic laboratories or commercially has facilitated efficient validation of findings from in vitro miRNA studies. Other animal models including nonhuman primate models have also been successfully used for the preclinical safety and toxicology testing of miRNA therapeutics supporting the initiation of several human miRNA therapeutic clinical trials.

Like other drug classes, the development of miRNA-based drugs must go through a sequence of developmental stages, spanning from discovery to preclinical studies, toxicology assessment, pre-IND, and multiple phases of human clinical trials prior to approval by regulatory agencies before market entry (Figure 5) [142].

In the conventional process of drug development, the duration from identifying a drug target to drug discovery, lead development, preclinical and phase 1–3 human clinical trials, FDA approval, and subsequent phase 4 studies usually spans a number of years. However, the escalating costs and time required in this process have become unsustainable, urging the imperative to hasten drug discovery and development while curbing associated expenses and timeframes.

Conversely, leveraging RNA-based methodologies, particularly miRNA-based approaches, holds promise for expediting both the discovery and development of drugs, potentially mitigating attrition rates, reducing time constraints, and cutting costs.

## 9. Progress in Preclinical Validation of miRNA Therapeutics

As discussed in the literature [77,182], single miRNAs can not only regulate an entire pathway and multitude of mRNA targets but also each mRNA may also be targeted by multiple miRNAs [278]. It has been suggested that the broad regulatory reach of miRNAs emerges from their ability to bind to target mRNAs even without perfect pairing. This characteristic enables a single miRNA to regulate multiple targets involved in similar cellular processes, amplifying the cellular response. While a single miRNA might inhibit numerous genes, its impact on each gene tends to be moderate [115], and multiple miRNAs can regulate the expression of a single gene [116,278], further amplifying the cellular response. Because a single miRNA has the potential to bind to as many as 200 target mRNAs, each with diverse functions, including transcription factors, receptors and many others, consequently, entire signaling pathways can be regulated by a single miRNA [116] or miRNA clusters [279]. However, the role of miRNAs in the regulation of multiple genes and their extensive effects within regulatory networks pose significant challenges for miRNA therapeutics [280]. Therefore, the successful development of miRNA-based therapeutics requires a comprehensive functional characterization and validation of the molecular effects of each miRNA prior to their application as therapeutics [77,182].

As summarized in Figure 5, the functional characterization of each miRNA requires several essential steps, including confirming its disease relevance authenticity as a genuine miRNA, improving miRNA target prediction algorithms, and experimentally validating their impact on intended targets in relevant preclinical models [77].

## 10. Off-Target Effects of miRNA Therapeutics

Among the challenges inherent in miRNA-based therapeutics, off-target effects and associated toxicities emerge due to the ability of each miRNA to regulate the expression of multiple genes and are one of the main challenges associated with miRNA therapeutics. Thus, further research is essential for the development of miRNAs as effective cancer therapeutics. Moreover, several miRNAs exhibit dysregulation not just within cancer cells but also in other cells within the TME, potentially leading to opposing functions.

Additionally, depending on the route of administration and the mechanism enabling intracellular delivery, miRNA therapeutics might not be confined exclusively to the targeted tissue or cells, potentially resulting in systemic side effects [77].

For instance, MRX34, a synthetic miR-34a mimic acting as a tumor suppressor [263], was administered systemically using a liposomal amphoteric (pH-dependent) delivery method, which takes advantage of the low-pH environment of the TME [264]. This approach aimed to treat various solid tumors and hematologic malignancies (NCT01829971) but was terminated early due to severe immune-related adverse effects that resulted in the death of four patients [82,83].

Previous animal studies, however, demonstrated that a miR-34a mimic was not only taken up by tumor tissue but also by bone marrow and spleen [281,282], which are involved in the generation and preservation of immune cells. Supporting these preclinical observations, the clinical testing of the miR-34a mimic demonstrated a dose-dependent change in several target genes in white blood cells [83]. As a result, the miR-34a mimic, aside from its role as a tumor suppressor, also affects immune cells by modulating calcium or chemokine signaling, such as the CXCL10/CXCL11/CXCR3-axis in CD4+, CD8+ T cells, and M1 macrophages [283,284].

Despite the lack of a direct causal link between patient death and the function of miR-34a in immune cells, the miR-34a function in immune cells has not been yet established. However, the serious and deadly adverse effects of miR-34a mimic MRX34 underscores the need for a priori risk assessment of miRNA therapeutics, specifically their potential off-target effects in other unintended tissues, highlighting the need for the development of more precise tissue target delivery systems.

## 11. Challenges and Future Perspectives

In addition to their potential utility as biomarkers [68], miRNA mimics and inhibitors provide significant potential as therapeutics. Given that many miRNAs function as oncomirs or tumor suppressors, restoring dysregulated miRNA levels to those of healthy tissues might aid in maintaining the naturally occurring anti-tumor regulatory mechanisms.

While many new immunotherapeutic such as antibodies, recombinant proteins, cell therapies, and small molecules have shown their success in the treatment of various types of cancers [285], because of the global success of mRNA vaccines as modulators of immune stimulation for tackling the COVID-19 pandemic, there has been a resurgence of RNA-based cancer immunotherapies [225].

Moreover, further improvements in RNA chemistry and delivery methods are opening up new opportunities for RNA-based therapeutics. Although, drug development, including RNA-based drugs, often requires many years and substantial costs before approval by the FDA or other regulatory agencies (Figure 5), the global threat of the COVID-19 pandemic has catalyzed the extremely rapid development of a new class of mRNA-based vaccines. These mRNA vaccines, whether developed by BioNTech in partnership with Pfizer or Moderna, have shown to deliver on their promise, i.e., they can be developed extremely quickly, can be manufactured under GMP-compliant manufacturing processes, and can be scaled for rapid availability of large numbers of doses, are safe and are active at a relatively low dose range [140,225,286,287]. Because of this, many companies are now leveraging the experience gained from the COVID-19 vaccine development to develop RNA-based therapies for cancer and, potentially, for other diseases. This has, of course, reignited the potential of miRNAs as both diagnostics and therapeutics.

The approval of numerous antisense, small interfering RNA (siRNA), and mRNA-based therapeutics and mRNA vaccines has affirmed their potential and paved the way for their application in new medical indications.

The field of RNA-based therapies encompasses a diverse spectrum of research areas, spanning RNA engineering and chemistry, which involves different modifications to enhance stability and pharmacokinetics and minimize nonspecific adverse effects and advancements in delivery technologies.

Furthermore, novel RNA constructs including those self-amplifying RNAs, circular RNAs, siRNAs, as well as gene editing [Cas9 mRNA, single guide RNA (sgRNA)] all hold promise for next-generation cancer therapy.

However, despite this potential over many years, there are several challenges including sensitivity, specificity, selectivity, toxicity immunogenicity, and delivery, among many others, which are a significant barrier to exploiting the full potential of miRNAs as therapeutics.

In addition, the development of novel and improved targeted delivery systems is vital for the effective delivery of miRNAs to the targeted tissue or cells. The delivery method must be target-specific, and capable of delivering the miRNA therapeutics to the targeted tissue or cells (i.e., disease site) [142].

As illustrated in Figure 4, the future of RNA-based therapies hinges on targeted delivery mechanisms. These encompass lipid and polymer nanoparticles, cellular or extracellular vesicle packaging, hybrid systems, and viral vectors, all aiming to enhance the therapeutic effectiveness of these therapies while reducing potential side effects.

In addition, there are other challenges that need to be addressed with regard to the sensitivity, specificity, selectivity, toxicity, and clinical applicability of miRNAs as therapeutics and therapeutic targets. Because each miRNA regulates more than one gene, sometimes a single miRNA can potentially regulate multiple cellular pathways via interacting with multiple targets. This phenomenon is referred to as “too many targets for miRNA effect” (TMTME) [128]. Likewise, each mRNA is regulated by more than one miRNA [77]. While these characteristics of miRNAs position them as a potent therapeutic class, they also pose a significant challenge in managing adverse effects observed in clinical trials [77]. Because of this, Zhang et al. [128] proposed that adverse events observed in terminated clinical trials involving miRNA therapeutics could be attributed to the broad-ranging effects of miRNAs.

Moreover, the type of miRNAs can vary throughout the course and stages of cancer, making target prediction more complex. Yet, this variation could be advantageous in associating specific miRNAs with particular cancer stages. Therefore, novel approaches for predicting miRNA targets may be necessary to validate these predictions.

Additional issues that need to be resolved include immunogenic reactions. Although viral delivery systems undoubtedly enhance cellular uptake and expression of miRNAs, they are associated with various side effects, including immunogenicity [233]. To address this issue, there needs to be a better understanding of the prevalence of immunogenic reactions resulting from viral transfer systems. Specifically, it is crucial to clarify the extent to which immunogenic reactions occur due to viral transfer systems.

In addition to viral transfer systems, it is essential to investigate whether modifications to miRNAs, such as LNA miRNAs and artificial miRNAs (amiRNAs) [288], or the use of miRNA-interfering molecules like small cell-permeable molecules, various delivery systems including biodegradable 3D matrices, functionalized metals as carriers, biomolecule conjugates including aptamers, can lead to severe immunogenic reactions. Therefore, exploring the possibility of reducing immunogenic reactions by masking reactive components should be considered. An overview of the extent to which immunogenic reactions occur was provided in a recent study [77].

Moreover, as summarized in Table 3 and discussed in detail in the literature [77], there are many other key outstanding questions that must be addressed before miRNA therapeutics can become widely adopted as novel therapeutics in the clinic.

These findings emphasize the need for further investigation in developing miRNAs as both novel therapeutics and therapeutic targets for cancer.

## 12. Conclusions

Because many miRNAs are abnormally expressed in many cancers acting as oncogenes or a tumor suppressors [98,289,290,291], they have emerged as potential biomarkers [68], therapeutic targets, and therapeutics [81]. However, there are many outstanding issues and challenges with respect to sensitivity, specificity, selectivity, and associated off-targeting effects of miRNA therapeutics since each miRNA appears to regulate more than one target and each target is regulated more than one miRNA leading to undesired toxicity hence limiting their use as therapeutics.

Currently, the majority of the miRNA therapeutics are still in preclinical or early phases of human clinical trials; as such, it is awaited to see how other miRNA therapeutics perform in human clinical trials in terms of toxicity or side effects. Notably, a few recent clinical trials using miRNA therapeutics have reported some serious adverse events. For example, MRX34, a microRNA liposomal injection developed by Mirna Therapeutics, Inc. evaluated in a phase 1 clinical trial for its efficacy against melanoma, was withdrawn (NCT02862145) or terminated (NCT01829971) [82,83,84] due to serious adverse events. As a result, numerous challenges must be addressed to bring therapeutic miRNAs into clinical practice. These include establishing miRNA specificity, sensitivity, and selectivity to their intended targets, reducing immunogenic reactions and adverse events, determining optimal dosing for the desired therapeutic effect while minimizing side effects [77], and developing improved methods for targeted delivery.

Despite these significant challenges, the potential of miRNAs as a therapeutic approach for various diseases is clear. Further research will be necessary to establish whether miRNAs can effectively serve as therapeutics or therapeutic targets for clinical applications.

## Figures and Tables

**Figure 1 ijms-25-01469-f001:**
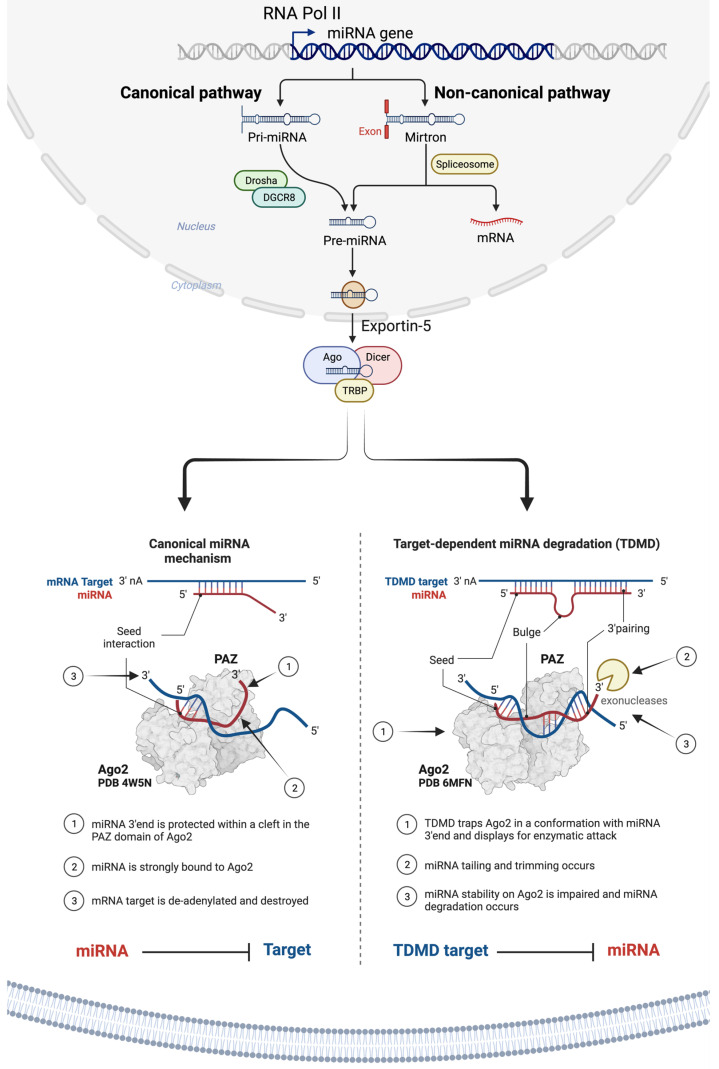
Illustration of miRNA biogenesis, processing, and mechanisms of translational suppression or degradation of target RNA. miRNAs are a class of small, single-stranded non-coding RNAs that function as a guide molecule in RNA silencing and hence modulate the expression of most mRNAs. The miRNA: target–mRNA interaction usually occurs at the 5′ end of the miRNA (i.e., ‘seed’ region). However, recent evidence suggests that there is a special class of target mRNAs which bind the miRNA not only through the ‘seed’ region, but also through a second region of complementarity at the 3′ end of the miRNA. The extended complementarity forces the miRNA out of Ago2, where it becomes accessible to enzymatic degradation. This phenomenon is referred to as the target-directed miRNA degradation mechanism (TDMD). Created with BioRender.com (accessed on 16 January 2024).

**Figure 2 ijms-25-01469-f002:**
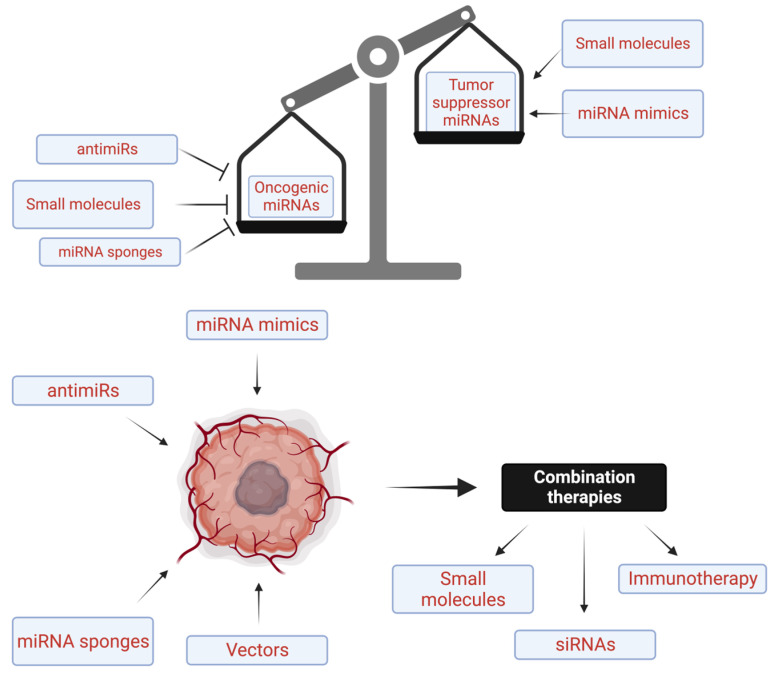
Schematic overview of miRNA therapeutic strategies to regulate the function of oncogenic and tumor suppressor miRNAs involved in cancer. (**Top Panel**): The strategy of miRNA therapeutics is based on restoring the balance of oncogenic miRNAs and tumor suppressor miRNAs. This involves downregulating the expression of oncomir RNAs (oncomirs) or upregulating the expression of tumor suppressor miRNAs. (**Bottom panel**): Therapeutic manipulations involving miRNAs can target the expression or function of pathologically significant miRNAs through various approaches. These methods include miRNA inhibitors (antagomirs or antimirs) that degrade or block the function of endogenous miRNAs, synthetic miRNA mimics that replicate endogenous miRNA functions, miRNAs expressed via viral vectors, small-molecule inhibitors that disrupt miRNA biogenesis, or miRNA sponges that functionally inhibit endogenous miRNAs by diverting them from their mRNA targets. Moreover, combining miRNAs with chemotherapies, immunotherapies, other traditional drugs or therapies, or siRNAs represents an additional strategy to counter drug resistance. Created with BioRender.com (accessed on 16 January 2024).

**Figure 3 ijms-25-01469-f003:**
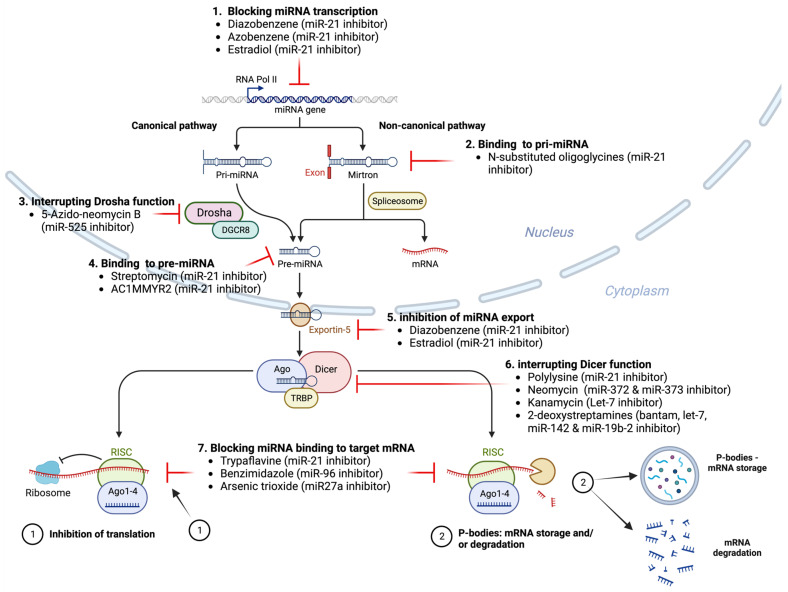
Diagram depicting various mechanisms employed by small-molecule inhibitors to target and inhibit specific miRNAs. “--|"represent the inhibition or blockage of a process. Created with BioRender.com (accessed on 16 January 2024).

**Figure 4 ijms-25-01469-f004:**
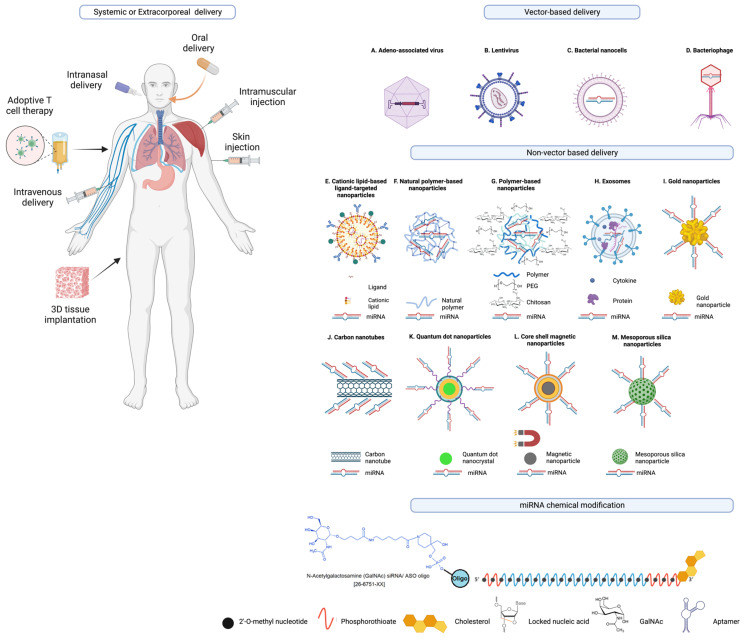
Examples of miRNA delivery systems. miRNA therapeutics can be administered orally or intranasally or through venous (intravenously) or muscle (intramuscularly) or skin (subcutaneously) injections, or via cell-/tissue-directed approaches, or adoptive cell transfer, or the implantation of 3D matrices that release miRNA therapeutics, or other extracorporeal miRNA delivery strategies. Other modes of delivery of miRNA therapeutics include vector based and non-vector-based delivery systems including (**A**) adeno-associated virus (**B**) Lentivirus; (**C**) bacterial nanocells; (**D**) bacteriophages; liposomes, including monovalent and multivalent lipids such as (**E**) cationic lipid-based ligand-targeted nanoparticles; (**F**) natural polymer-based nanoparticles; (**G**) polymer-based nanoparticles (natural, green and synthetic, blue) conjugated with polyethylene glycol (PEG); (**H**) extracellular vesicles or exosomes; (**I**) gold nanoparticles [224]; (**J**) carbon nanotubes; (**K**) quantum dot nanoparticles; (**L**) core–shell magnetic nanoparticles; (**M**) mesoporous silica nananoparticles and others such as polymeric micelles, and mesoporous silica nanoparticles are the examples of nanocarriers as drug-delivery systems. Moreover, there have been efforts to improve the serum stability, pharmacokinetics, and tissue specificity by targeted delivery of miRNA mimics, miRNA inhibitors, and other nucleic acid therapeutics through the incorporation of various chemical modifications and/or conjugation of these RNA and nucleic acid therapeutics to biomolecules to facilitate receptor-mediated uptake such as N-acetylgalactosamine (GalNAc), 2′-O-methyl nucleotide, phosphorothioate, cholesterol, locked nucleic acid (LNA), and aptamer moieties. Created with BioRender.com (accessed on 16 January 2024).

**Figure 5 ijms-25-01469-f005:**
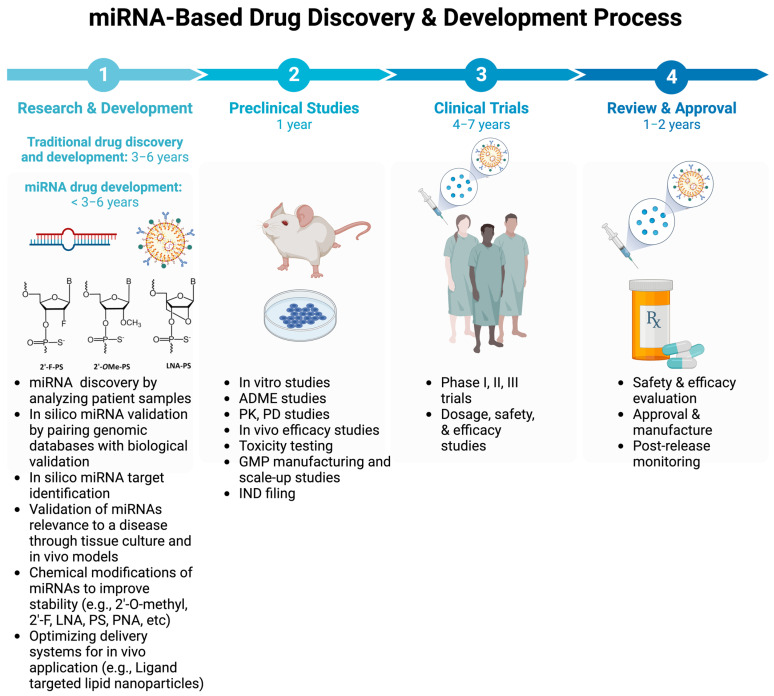
Illustration of the miRNA-based drug discovery and development process beginning from target identification and miRNA discovery to FDA-approved miRNA therapeutics on the market. In the traditional drug development process, the timeline from target identification and drug discovery to phase 1–3 human clinical trials and, ultimately, FDA approval, followed by phase 4 studies, can go on for several years. Conversely, RNA-based and, more specifically, miRNA-based drug discovery and development can potentially mitigate attrition rates, time constraints, and costs. The initial phase in developing miRNA therapeutics involves systematically selecting potential miRNA candidates by analyzing patient samples and validating their relevance to a particular disease of interest through tissue culture and in vivo models. Various publicly available genomic and proteomic databases from diverse healthy and diseased tissues can aid in identifying promising miRNA candidates when combined with biological validation. The next step often involves modifying miRNA therapeutics and optimizing delivery systems suitable for in vivo applications. A major concern with miRNA therapeutics is their susceptibility to degradation by nucleases and endosomal escape. To improve the stability of miRNA therapeutics, chemical modifications such as adding a 2′-O-methyl group, 2-F group, locked nucleic acids (LNAs), or peptide nucleic acids (PNAs) as well as a phosphorothioate group considerably enhance stability. Various encapsulation methods such as lipid nanoparticles, neutral lipid emulsions, or dendrimer complexes equipped with a targeting moiety have been employed for improved delivery to target tissue and disease sites. Yet, challenges remain in transitioning these delivery systems into clinical applications due to potential immune activation effects and the lack of precise targeting for disease sites. Successful translation of lead miRNAs into clinical studies requires rigorous disease-specific in vivo testing using rodent and non-human primate models. Rigorous evaluation of toxicity data and target engagement is crucial to avert early setbacks in clinical trials. Created with BioRender.com (accessed on 16 January 2024).

**Table 1 ijms-25-01469-t001:** Examples of miRNA-based therapeutics in the preclinical development phase for human malignancies.

Therapeutic Molecule	Target miRNA	Disease	Biopharmaceutical Company	Stage of Development
RG-012	miRNA-21	Alport nephropathy	Regulus therapeutics (with the strategic alliance with Genzyme)	Preclinical stage
MGN-1374	miRNA-15 and miR-195	Post-myocardial infarction	miRagen therapeutics	Preclinical stage
MGN-2677	miR-143/145	Vascular disease	miRagen therapeutics	Preclinical stage
MGN-4220	miR-29	Cardiac fibrosis	miRagen therapeutics	Preclinical stage
MGN-4893	miR-451	For the treatment of disorders like abnormal red blood cell production	miRagen therapeutics.	Preclinical stage
MGN-5804	miR-378	Cardiometabolic disease	miRagen therapeutics	Preclinical stage
MGN-6114	miR-92	Peripheral arterial disease	miRagen therapeutics	Preclinical stage
MGN-9103	miR-208	Chronic heart failure	miRagen therapeutics	Preclinical stage
MRG-107	miR-155	Amyotrophic lateral sclerosis (ALS)	miRagen therapeutics	Completed preclinical stage

**Table 2 ijms-25-01469-t002:** Clinical trials with miRNA therapeutics for various diseases. NCT numbered trials are registered at ClinicalTrials.gov; EudraCT numbered trials are registered at EU Clinical Trials Register (clinicaltrialsregister.eu).

miRNA Drug Name	Targeted miRNA	Study Title	Mode of Action	Disease/Condition	Mode of Delivery	Phase	Status	Clinical Trial Number(s)	References
miR-10b	miR-10b	Evaluating the Expression Levels of MicroRNA-10b in Patients with Gliomas	miR-10b as diagnostic and in vitro testing of anti-mir-10b as therapeutic	Astrocytoma Oligodendroglioma Oligoastrocytoma Anaplastic Astrocytoma Anaplastic Oligodendroglioma Anaplastic Oligoastrocytoma Glioblastoma Brain Tumors Brain Cancer		Observational	Recruiting	NCT01849952	
INT-1B3	miR-193a-3p mimic	First-in-Human Study of INT-1B3 in Patients with Advanced Solid Tumors	miRNA mimic	Advanced solid tumors		Phase I	Recruiting	NCT04675996	NA
AMT-130	Artificial miRNA	Safety and Proof-of-Concept (POC) Study with AMT-130 in Adults with Early Manifest Huntington’s Disease	A miRNA expression	Huntington disease	Stereotaxic infusion/viral transfer (adeno-associated vector)	Phase I	Ongoing	NCT04120493	[146,147,148]
RG-012/lademirsen/SAR339375	miR-21	A Study of RG-012 in Subjects with Alport Syndrome	Anti-miR-21 Lademirsen—also known as RG-012, RG456070 or (SAR339375)	Alport syndrome	Subcutaneous injection/chemical modification (phosphorothioate)	Phase II	Completed	NCT03373786	[149,150,151]
RG-012/lademirsen/SAR339375	miR-21	Study of Lademirsen (SAR339375) in Patients with Alport Syndrome	Anti-miR-21 Lademirsen—also known as RG-012, RG456070 or (SAR339375)	Alport syndrome	Subcutaneous injection/chemical modification (phosphorothioate)	Phase II	Terminated	NCT02855268	[149,150,151]
RGLS4326	miR-17	A Study of RGLS4326 in Patients with Autosomal Dominant Polycystic Kidney Disease	Anti-miR-17	Autosomal dominant polycystic kidney disease	Administered via subcutaneous injection	Phase I	Completed	NCT04536688	
RG-125/AZD4076	miR-103/107	A Study to Assess the Safety and Tolerability of Single Doses of AZD4076 in Healthy Male Subjects	Anti-miR	Non-alcoholic Steatohepatitis (NASH)	Subcutaneous injection/biomolecule conjugation (GalNAc)	Phase I	Active, not recruiting	NCT02612662, NCT02826525	[150,152,153]
RG-125/AZD4076	miR-103/107	AZD4076 in Type 2 Diabetic Subjects with Non-Alcoholic Fatty Liver Disease	Anti-miR	T2DM With NAFLD	Subcutaneous injection/biomolecule conjugation (GalNAc)	Phase I	Completed	NCT02826525	[150,152,153]
MRG-110	miR-92a	Safety, Tolerability, Pharmacokinetics, and Pharmacodynamics of MRG-110 following Intradermal Injection in Healthy Volunteers	Anti-miR	Healthy volunteer	Skin injection/chemical modification (LNA)	Phase I	Completed	NCT03603431	[154,155]
MesomiR 1	miR-16	MesomiR 1: A Phase I Study of TargomiRs as 2nd or 3rd Line Treatment for Patients with Recurrent MPM and NSCLC	miRNA mimic	Malignant pleural mesothelioma, non–small cell lung cancer	Intravenously/vehicle transfer of nonliving bacterial nanocells (EDVs or TargomiRs)	Phase I	Completed	NCT02369198	[156,157,158]
CDR132L	miR-132	Clinical Study to Assess Safety, PK and PD Parameters of CDR132L	Anti-miR	Heart failure	Intravenously/chemical modification (LNA)	Phase I	Completed	NCT04045405	[159,160]
Remlarsen/MRG-201	miR-29	Efficacy, Safety, and Tolerability of Remlarsen (MRG-201) following Intradermal Injection in Subjects With a History of Keloids	miRNA mimic	Keloid disorder	Skin injection/biomolecule conjugation (cholesterol)	Phase II	Completed	NCT03601052	[161,162,163]
Miravirsen/SPC3649	miR-122	Long-Term Extension Study of Miravirsen among Participants with Genotype 1 Chronic Hepatitis C (CHC) Who Have Not Responded to Pegylated-Interferon Alpha Plus Ribavirin	Anti-miR	Chronic hepatitis C virus	Subcutaneous injection/chemical modification (LNA)	Phase II Phase II Phase II Phase II Phase II Phase I	Completed Completed Completed Unknown Unknown Completed	NCT02508090 NCT02508090, NCT02452814, NCT01200420, NCT01872936, NCT01727934, NCT01646489	[164,165,166,167,168]
Miravirsen/SPC3649	miR-122	Long Term Extension Study is Designed to Monitor Long-Term Efficacy and Safety of Miravirsen Sodium in Combination with Telaprevir and Ribavirin in Subjects with Chronic Hepatitis C Virus Genotype 1 Infection	Anti-miR	Chronic hepatitis C virus	Subcutaneous injection/chemical modification (LNA)	Phase II	Completed	NCT02452814	[164,165,166,167,168]
Miravirsen/SPC3649	miR-122	Multiple Ascending Dose Study of Miravirsen in Treatment-Naïve Chronic Hepatitis C Subjects	Anti-miR	Chronic hepatitis C virus	Subcutaneous injection/chemical modification (LNA)	Phase II Phase II Phase II Phase I	Completed Unknown Unknown Completed	NCT01200420	[164,165,166,167,168]
Miravirsen/SPC3649	miR-122	Miravirsen in Combination with Telaprevir and Ribavirin in Null Responder to Pegylated-Interferon Alpha Plus Ribavirin Subjects with Chronic Hepatitis C Virus Infection	Anti-miR	Chronic hepatitis C virus	Subcutaneous injection/chemical modification (LNA)	Phase II Phase II Phase I	Unknown	NCT01872936	[164,165,166,167,168]
Miravirsen/SPC3649	miR-122	Miravirsen Study in Null Responder to Pegylated Interferon Alpha Plus Ribavirin Subjects with Chronic Hepatitis C	Anti-miR	Chronic hepatitis C virus	Subcutaneous injection/chemical modification (LNA)	Phase II Phase II Phase I	Unknown	NCT01727934	[164,165,166,167,168]
Miravirsen/SPC3649	miR-122	Drug Interaction Study to Assess the Effect of Co-Administered Miravirsen and Telaprevir in Healthy Subjects	Anti-miR	Chronic hepatitis C virus	Subcutaneous injection/chemical modification (LNA)	Phase II Phase II Phase I	Completed	NCT01646489	[164,165,166,167,168]
RG-101	miR-122	A Randomized, Multi-Center, Phase 2 Study to Evaluate Safety and Efficacy of Subcutaneous Injections of RG-101 in Combination with Oral Agents in Treatment Naïve, Genotype 1 and 4, Chronic Hepatitis.	Anti-miR	Chronic hepatitis C virus	Subcutaneous injection/biomolecule conjugation (GalNAc)	Phase II		EudraCT numbers 2015-001535-21, 2015-004702-42, 2016-002069-77	[150,169,170]
RG-101	miR-122	A Multi-Center, Parallel Group, Open-Label, Phase 2 Study to Evaluate the Efficacy and Safety of a Single Subcutaneous Injection of RG-101 Combined with Oral GSK2878175	Anti-miR	Chronic hepatitis C virus	Subcutaneous injection/biomolecule conjugation (GalNAc)	Phase II		EudraCT numbers 2015-004702-42	[150,169,170]
RG-101	miR-122	An Observational Long-Term Safety and Efficacy Follow-Up Study of Subjects Who Have Previously Received RG-101	Anti-miR	Chronic hepatitis C virus	Subcutaneous injection/biomolecule conjugation (GalNAc)	Observational	Unknown	EudraCT numbers 2016-002069-77	[150,169,170]
MRX34	miR-34a	A Multicenter Phase I Study of MRX34, MicroRNA miR-RX34 Liposomal Injection	miRNA mimic	Primary liver cancer SCLC Lymphoma Melanoma Multiple myeloma Renal cell carcinoma NSCLC	Intravenously/vehicle transfer (liposomal)	Phase I	Terminated (5 immune related serious adverse events)	NCT01829971	[65,82,83]
MRX34	miR-34a	Pharmacodynamics Study of MRX34, MicroRNA Liposomal Injection in Melanoma Patients with Biopsy Accessible Lesions	miRNA mimic	Solid tumors (e.g., hepatocellular carcinoma, melanoma, SCLC, NSCLC, lymphoma, multiple myeloma, renal cell carcinoma)	Intravenously/vehicle transfer (liposomal)	Phase I Phase II	Withdrawn (5 immune related serious adverse events in Phase I)	NCT02862145	[65,82,83]
Cobomarsen/MRG-106	miR-155		Anti-miR	Mycosis fungoides (MF) Cutaneous T-cell Lymphoma (CTCL) Chronic Lymphocytic Leukemia (CLL) Diffuse large B-cell Lymphoma (DLBCL), ABC subtype Adult T-cell leukemia/lymphoma (ATLL)	Intravenously/chemical modification (LNA)	Phase I Phase II Phase II	Completed Terminated Terminated	NCT02580552, NCT03713320, NCT03837457	[171,172,173]
Serum MicroRNA-25	miR-25		Serum miR-25 as diagnostic	Pancreatic cancer	Serum miR-25	Observational	Not yet recruiting	NCT03432624	
Patisiran (ALN-TTR02),			RNAi therapeutic	Transthyretin (TTR)- mediated amyloidosis	ALN-TTR02 administered by intravenous infusion	Phase III	Completed	NCT01960348	[174]
miR-10	miR-10	Evaluating the Expression Levels of MicroRNA-10b in Patients with Gliomas	anti-miR-10	Glioma	Evaluating the expression levels of microRNA-10b in patients with gliomas	Observational	Recruiting	NCT01849952	

**Table 3 ijms-25-01469-t003:** Critical inquiries that must be addressed before clinical application of miRNA therapeutics.

1	What methods can be used to effectively guide therapeutic miRNAs/miRNA inhibitors to their intended target tissue and cells in vivo?
2	How can the design of miRNA/miRNA-based drugs and delivery vehicles be optimized to reduce or, ideally, eliminate unintended impacts on non-targeted cells?
3	What other strategies can be used to improve more accurate targeting for miRNA/miRNA inhibitor therapeutics?
4	Is there a risk of incompatibilities when using diverse carrier materials for advanced miRNA/miRNA inhibitor-based drug delivery, which may lead to undesired interactions between the materials and miRNA therapeutics?
5	Is there a risk of incompatibilities when using miRNA/miRNA inhibitor therapeutics in combination with traditional drugs pose the risk of incompatibilities?
6	Do modifications of miRNA or miRNA inhibitors such as LNA miRNA mimic or miRNA inhibitors and other amiRNAs, cell-permeable molecules, delivery methods including biodegradable 3D matrices, nanocarriers like functionalized metals, viral vector-based transfer systems, or biomolecule conjugate combinations such as aptamers invoke immunogenic responses? If so, can the activation of immunogenic responses be ameliorated through the masking of reactive components or moieties?
7	What is the level of risk associated with genomic integrations of viral transduction constructs that carry miRNA or miRNA inhibitors?
8	What is the impact of the expression of endogenous miRNAs and mRNAs on exogenously delivered therapeutic miRNAs and miRNA inhibitors which may be also affected by factors like cell type, cell cycle, and the cellular environment?
9	What is the necessary dosage for particular administration techniques for miRNA mimics or miRNA inhibitors, such as skin injection, infusion, or inhalation, and for nanocarriers such as biodegradable 3D matrices?
10	How can the administration of miRNA/miRNA inhibitor therapeutic doses be regulated along intricate in vivo delivery pathways?
11	Is it possible to achieve consistent and sustainable rates of cellular uptake of miRNA/miRNA inhibitor therapeutics under varying in vivo conditions?
12	In what ways can dosing of miRNA mimics and inhibitors support the desired gene targeting outcome?

## Data Availability

The data presented in this study are openly available in https://seer.cancer.gov, https://pubmed.ncbi.nlm.nih.gov/, https://clinicaltrials.gov/, and https://www.genomicseducation.hee.nhs.uk/genotes/knowledge-hub/non-coding-dna, all accessed on 26 July 2023.

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
