# Peer review of "Trials and Tribulations of MicroRNA Therapeutics"

_ijms, 2024, doi:10.3390/ijms25031469_

Round 1

Reviewer 1 Report

Comments and Suggestions for Authors

In this article, the author systematically reviewed the regulatory role of miRNAs in disease and their potentials as therapeutic targets. Clinical trials involving mRNA, miRNA and small molecule modulators of miRNA expression were briefly introduced and comprehensively summarized, providing a landscape of current status of RNA therapeutics. As particularly important parts of RNA therapeutics, developments in miRNA delivery and stabilization were also discussed. Other crucial challenges, including miRNA target validation, off-target effects and toxicities were summarized and discussed.

This review covered miRNA therapeutics comprehensively. It should be very useful for readers interested in this area. It can be published after resolving the following issues.

Minor issues:

1.     Abstract. Over 2/3 of the content was talking about the regulatory function of miRNA and their involvement in diseases while this review is on miRNA therapeutics. The abstract should be revised to reflect the major topic.

2.     Line 177 referred to Figure X, which could not be found. Please fix it.

3.     Section 5.2 is in chaos. E.g., Line 191-197 should be integrated into one paragraph as well as line 198-205.

4.     Please check the format throughout the article. Numerous lines were highlighted in grey.

5.     Figure 5, the top line of flow chart is not aligned.

6.     In line 130, “MiRNAs” should be corrected as “miRNAs”.

7.     It seems that the order of Figure 2 and Figure 3 is reversed.

8.     In the manuscript, the order of the tables is confused: Table 2, Table 3, Table 1.

9.     In line 227, there is an extra “=” at the end of the sentence.

10.  In line 537-539the author mentioned “more innovative approaches for predicting miRNA targets might be required to validate the predicted targets”. It is desired to give a more detailed information about these methods, for example: the deficiencies of existing methods and possible directions of future methods. The same is for line 543-544. More detailed information about “better understanding” is desired.

Comments on the Quality of English Language

Minor editing of English language is required.

Author Response

Comments and Suggestions for Authors

In this article, the author systematically reviewed the regulatory role of miRNAs in disease and their potentials as therapeutic targets. Clinical trials involving mRNA, miRNA and small molecule modulators of miRNA expression were briefly introduced and comprehensively summarized, providing a landscape of current status of RNA therapeutics. As particularly important parts of RNA therapeutics, developments in miRNA delivery and stabilization were also discussed. Other crucial challenges, including miRNA target validation, off-target effects and toxicities were summarized and discussed.

This review covered miRNA therapeutics comprehensively. It should be very useful for readers interested in this area. It can be published after resolving the following issues.

Response to reviewer 1: The author thanks the reviewer for taking the time to review this manuscript and for the comments which were very helpful in revising the manuscript to convey the message more clearly. Below is a point-by-point response to each comment. The corresponding revisions/corrections highlighted/in track changes in the re-submitted files.

Minor issues:

  1. Abstract. Over 2/3 of the content was talking about the regulatory function of miRNA and their involvement in diseases while this review is on miRNA therapeutics. The abstract should be revised to reflect the major topic.

Response: The author thanks to the reviewer for the important critique and helpful comments. The abstract has been revised accordingly to reflect the major topic.

  1. 2.     Line 177 referred to Figure X, which could not be found. Please fix it.

Response: Figure X on Line 177 cannot be found but the masuncript was searched thoroughly for Figure X and none could be found.

  1. Section 5.2 is in chaos. E.g., Line 191-197 should be integrated into one paragraph as well as line 198-205. 

Response: Section 5.2 has been revised.

  1. Please check the format throughout the article. Numerous lines were highlighted in grey.

Response: Corrected.

  1. Figure 5, the top line of flow chart is not aligned.

Response: Corrected.

  1. In line 130, (104) “MiRNAs” should be corrected as “miRNAs”.

Response: Corrected.

  1. It seems that the order of Figure 2 and Figure 3 is reversed.

Response: The order of the figures has been corrected.

  1. In the manuscript, the order of the tables is confused: Table 2, Table 3, Table 1.

Response: The order of the tables is now accurate.

  1. In line 227, there is an extra “=” at the end of the sentence.

Response: miR- corrected as miR-21.

  1. In line 537-539,the author mentioned “more innovative approaches for predicting miRNA targets might be required to validate the predicted targets”. It is desired to give a more detailed information about these methods, for example: the deficiencies of existing methods and possible directions of future methods. The same is for line 543-544. More detailed information about “better understanding” is desired.

Response: More detailed information on the more innovative approaches for predicting miRNA targets have been provided. In addition, more detailed information about “better understanding” on the prevalence of immunogenic reactions have been provided.

Comments on the Quality of English Language: Minor editing of English language is required.

Submission Date: 22 December 2023

Date of this review: 05 Jan 2024 10:13:09

Reviewer 2 Report

Comments and Suggestions for Authors

This is a welcome summary of microRNA therapeutics, which focuses on trials and tribulations of microRNAs therapeutics use. More specifically, the review tries to give a perspective on the biochemical functions of microRNAs involved in keeping body homeostasis and their disruption in disease and possible strategies to microRNA delivery.

The author, after an accurate exploration of the literature, provides detailed information on this topic.

Minor comments:

In Figure 1, please mark the correspondence numbers 1,2,3 (numbers in the circles) to identify pathways. Same thing for figure 2 with 1 and 2 (numbers in the circles).

The sentences on line 38, from line 246 to line 250, should follow the journal style, also in lines 380 and 403.

Author Response

Comments and Suggestions for Authors

This is a welcome summary of microRNA therapeutics, which focuses on trials and tribulations of microRNAs therapeutics use. More specifically, the review tries to give a perspective on the biochemical functions of microRNAs involved in keeping body homeostasis and their disruption in disease and possible strategies to microRNA delivery.

The author, after an accurate exploration of the literature, provides detailed information on this topic.

Response to reviewer 2: The author thanks the reviewer for taking the time to review this manuscript and for the comments which were very helpful in revising the manuscript to convey the message more clearly. Below is a point-by-point response to each comment. The corresponding revisions/corrections highlighted/in track changes in the re-submitted files.

Minor comments:

Comments: In Figure 1, please mark the correspondence numbers 1,2,3 (numbers in the circles) to identify pathways. Same thing for figure 2 with 1 and 2 (numbers in the circles).

Response: Both Figure 1 and 2 have been corrected to reflect the relevant correspondence numbers 1,2,3 (Figure 1), and 1, 2 (Figure 2), respectively.

Comments: The sentences on line 38, from line 246 to line 250, should follow the journal style, also in lines 380 and 403.

Response: Font style have been corrected according to the journal font style (Palatino Linotype).

Submission Date: 22 December 2023 

Date of this review: 03 Jan 2024 15:50:32

Reviewer 3 Report

Comments and Suggestions for Authors

The manuscript titled "Trials and tribulations of microRNA therapeutics" presents a comprehensive review of the molecular pathways involved in pathogenesis with some insights about the therapeutic approaches. The manuscript delves into a novel and significant subject matter that has received considerable attention and holds practical relevance. In conclusion, the article authored by Attila A. Seyhan is deemed suitable for publication.

Author Response

Comments and Suggestions for Authors

The manuscript titled "Trials and tribulations of microRNA therapeutics" presents a comprehensive review of the molecular pathways involved in pathogenesis with some insights about the therapeutic approaches. The manuscript delves into a novel and significant subject matter that has received considerable attention and holds practical relevance. In conclusion, the article authored by Attila A. Seyhan is deemed suitable for publication.

Response to reviewer 3: The author expresses gratitude to the reviewer for dedicating time to review the manuscript and for the positive feedback.

Submission Date: 22 December 2023

Date of this review: 13 Jan 2024 17:58:10

Reviewer 4 Report

Comments and Suggestions for Authors

The author presented a very detailed review of the R&D landscape of miRNA as therapeutics. Here are a few comments for the authors to improve the manuscript.

1.      The content in lines 117-122 is the same as in lines 123-128. The author should check this.

2.      The figures are out of order (1,3,2,4,5).

3.      Line 445: when discussing computational methods that were used in miRNA research, the authors should consider mechanistic modeling studies (e.g. models that simulate miRNA-mediated pathways, see two in-depth reviews PMID-27317695 and 30669429) in addition to the classical bioinformatics tools.

Author Response

Comments and Suggestions for Authors

The author presented a very detailed review of the R&D landscape of miRNA as therapeutics.

Here are a few comments for the authors to improve the manuscript.

Response to reviewer 4: The author thanks the reviewer for taking the time to review this manuscript and for the comments which were very helpful in revising the manuscript to convey the message more clearly. Below is a point-by-point response to each comment. The corresponding revisions/corrections highlighted/in track changes in the re-submitted files.

  1. The content in lines 117-122 is the same as in lines 123-128. The author should check this.

Response: Duplicated sentence has been deleted (page 3, now lines 113-118).

  1. The figures are out of order (1,3,2,4,5).

Response: The order of the figures has been corrected.

  1. Line 445: when discussing computational methods that were used in miRNA research, the authors should consider mechanistic modeling studies (e.g. models that simulate miRNA-mediated pathways, see two in-depth reviews PMID-27317695 and 30669429) in addition to the classical bioinformatics tools.

Response: The author thanks to the reviewer for the important and helpful comments. Mechanistic modeling studies (e.g. models that simulate miRNA-mediated pathways) with relevant citations to the review articles (PMID-27317695 and 30669429) have been provided in the relevant section of the manuscript.

Submission Date: 22 December 2023

Date of this review: 10 Jan 2024 11:51:44
